# A computational lens into how music characterizes genre in film

**Benjamin Ma**[1]☯*, **Timothy Greer**[1]☯*, **Dillon Knox**[1]☯*, **Shrikanth Narayanan**[1,2]

**1** Department of Computer Science, University of Southern California, Los Angeles, CA, United States of America, **2** Department of Electrical and Computer Engineering, University of Southern California, Los Angeles, CA, United States of America

☯ These authors contributed equally to this work.
* benjamjm@usc.edu (BM); timothdg@usc.edu (TG); dillonkn@usc.edu (DK)

**Data Availability Statement:** All relevant data are within the paper and its Supporting information files.

**Funding:** The study was done at the Center for Computational Media Intelligence at USC, which is

## Abstract

Film music varies tremendously across genre in order to bring about different responses in an audience. For instance, composers may evoke passion in a romantic scene with lush string passages or inspire fear throughout horror films with inharmonious drones. This study investigates such phenomena through a quantitative evaluation of music that is associated with different film genres. We construct supervised neural network models with various pooling mechanisms to predict a film's genre from its soundtrack. We use these models to compare handcrafted music information retrieval (MIR) features against VGGish audio embedding features, finding similar performance with the top-performing architectures. We examine the best-performing MIR feature model through permutation feature importance (PFI), determining that mel-frequency cepstral coefficient (MFCC) and tonal features are most indicative of musical differences between genres. We investigate the interaction between musical and visual features with a cross-modal analysis, and do not find compelling evidence that music characteristic of a certain genre implies low-level visual features associated with that genre. Furthermore, we provide software code to replicate this study at https://github.com/usc-sail/mica-music-in-media. This work adds to our understanding of music's use in multi-modal contexts and offers the potential for future inquiry into human affective experiences.

## Introduction

Music plays a crucial role in the experience and enjoyment of film. While the narrative of movie scenes may be driven by non-musical audio and visual information, a film's music carries a significant impact on audience interpretation of the director's intent and style [1]. Musical moments may complement the visual information in a film; other times, they flout the affect conveyed in film's other modalities (e.g.—visual, linguistic). In every case, however, music influences a viewer's experience in consuming cinema's complex, multi-modal stimuli. Analyzing how these media interact can provide filmmakers and composers insight into how to create particular holistic cinema-watching experiences.

supported by a research award from Google. The
funders had no role in study design, data collection
and analysis, decision to publish, or preparation of
the manuscript.

**Competing interests:** Funding from Google does
not alter our adherence to PLOS ONE policies on
sharing data and materials of this paper. The
authors have declared that no other competing
interests exist related to this effort.

We hypothesize that musical properties, such as timbre, pitch, and rhythm, achieve particular stylistic effects in film, and are reflected in the display and experience of a film's accompanying visual cues, as well as its overall genre classification. In this study, we characterize differences among movies of different genres based on their film music scores. While this paper focuses on how music is used to support specific cinematic genres, created to engender particular film-watching experiences, this work can be extended to study other multi-modal content experiences, such as viewing television, advertisements, trailers, documentaries, music videos and musical theatre.

## Related work

### Music use across film genre

Several studies have explored music use in cinema. Music has been such an integral part of the film-watching experience that guides for creating music for movies have existed since the Silent Film era of the early 20th century [2]. Gorbman [3] noted that music in film acts as a signifier of emotion while providing referential and narrative cues, while Rodman [4] points out that these cues can be discreetly "felt" or overtly "heard." That stylistic musical effects and their purpose in film is well-attested provides an opportunity to study how these musical structures are used.

Previous work has made preliminary progress in this direction. Brownrigg presented a qualitative study on how music is used in different film genres [5]. He hypothesized that film genres have distinctive musical paradigms existing in tension with one another. By this token, the conventional score associated with one genre can appear in a "transplanted" scene in another genre. As an example, a Science Fiction movie may use musical conventions associated with Romance to help drive the narrative of a subplot that relates to love. In this paper, we use a multiple instance machine learning approach to study how film music may provide narrative support to scenes steeped in other film genres.

Other studies have taken a more quantitative approach, extracting audio from movies to identify affective content [6, 7]. Gillick analyzed soundtracks from over 40,000 movies and television shows, extracting song information and audio features such as tempo, danceability, instrumentalness, and acousticness, and found that a majority of these audio features were statistically significant predictors of genre, suggesting that studying music in film can offer insights into how a movie will be perceived by its audience [8]. In this work, we use musical features and state-of-the-art neural embeddings to study film genre.

Another study that used machine learning techniques, by Austin et al., found timbral features most discriminatory in separating movie genres [1]. In prior work, soundtracks were analyzed without accounting for *if* or *for how long* the songs were used in a film. We extend these studies by investigating how timestamped musical clips that are explicitly used in a film relate to that film's genre.

### Musical-visual cross-modal analysis

Previous research has established a strong connection between the visual and musical modes as partners in delivering a comprehensive narrative experience to the viewer [9–12]. Cohen [10] argued that music "is one of the strongest sources of emotion in film" because it allows the viewer to subconsciously attach emotional associations to the visuals presented onscreen. Wingstedt [13] advanced this theory by proposing that music serves not only an "emotive" function, but also a "descriptive" function, which allows the soundtrack to describe the setting of the story-world (e.g., by using folk instruments for a Western setting). In combination with

its emotive function, music's descriptive function is critical in supporting (or undermining) the film genre characterized by the visuals of the film.

In this study, we use screen brightness and contrast as two low-level visual features to describe the visual mode of the film. Chen [14] found that different film genres have characteristically different average brightness and contrast values: Comedy and Romance films have higher contrast and brightness, while Horror, Sci-Fi, and Action films were visually darker with less contrast. Tarvainen [15] established statistically significant correlations between brightness and color saturation with feelings of "beauty" and "pleasantness" in film viewers, while darkness and lack of color were associated with "ugliness" and "unpleasantness." This result is complementary to Chen's finding: Comedy and Romance films tend to evoke "beauty" and "pleasantness," while Action, Horror, and Sci-Fi tend to emphasize gritty, muddled, or even "unpleasant" and "ugly" emotions.

## Multiple instance learning

Multiple instance learning (MIL) is a supervised machine learning method where ground truth labels are not available for every instance; instead, labels are provided for sets of instances, called "bags." The goal of classification in this paradigm is to predict bag-level labels from information spread over instances. In our study, we treat each of the 110 films in the dataset as a bag, and each musical cue within the film as an instance. A musical cue is a single timestamped instance of a track from the soundtrack that plays in the film.

Strong assumptions about the relationship between bags and instances are common, including the standard multiple instance (MI) assumption where a bag (movie) contains a label if and only if there exists at least one instance (a cue within that movie) that is tagged with that label. In this work, we make the *soft bag* assumption, which allows for a negative-labeled bag to contain positive instances [16]. In other words, a film can contain musical moments characteristic of genres that are outside its own.

**Simple MI.** *Simple MI* is a MI method in which a summarization function is applied to all instances within a bag, resulting in a single feature vector for the entire bag. Then, any number of classification algorithms can be applied to the resulting single instance classification problem. Here, the arithmetic mean is used as a straightforward summarization function, as applied in [17].

**Instance majority voting.** In *instance majority voting*, each instance within a given bag is naïvely assigned the labels of that bag, and a classifier is trained on all instances. Bag-level labels are then assigned during inference using an aggregation scheme, such as majority voting [18]. As an example, a movie that is labeled as a "comedy" would propagate that label to the cues it contains during model training, and then majority voting across cues would be used to predict the final label for the movie during inference.

**Neural network approaches.** Neural network approaches within an MIL framework have been used extensively for sound event detection (SED) tasks with weak labeling. Ilse et al. [19] proposed an attention mechanism over instances and demonstrated competitive performance on several benchmark MIL datasets. Wang et al. [20] compared the performance of five MIL pooling functions, including attention, and found that linear softmax pooling produced the best results. Kong et al. [18] proposed a new feature-level attention mechanism, where attention is applied to the hidden layers of a neural network. Gururani et al. [21] used an attention pooling model for a musical instrument recognition task, and found improved performance over other architectures, including recurrent neural networks. In this work, we compare each of these approaches for the task of predicting a film's genre from its music.

## Contribution of this work

In this work, we objectively examine the effect of musical features on perception of film. We curate and release a dataset of processed features from 110 popular films and soundtracks, and share the code we use for our experiments (https://github.com/usc-sail/mica-music-in-media). To our knowledge, this is the first study that applies deep learning models on musical features to predict a film's genre. Additionally, we interpret these models via a permutation feature importance analysis on MIR features. This analysis suggests which interpretable musical features are most predictive of each film genre studied. Lastly, we conduct a novel investigation on the interaction between the musical and low-level visual features of film, finding that musical and visual modes may exhibit characteristics of different genres in the same film clips. We believe that this work also sets the foundation that can be extended to help us better understand music's role as a significant and interactive cinematic device, and how viewers respond to the cinematic experience.

## Research data collection and curation

### Film and soundtrack collection

**Soundtracks.** We collected the highest-grossing movies from 2014-2019 in-house (boxofficemojo.com). We identified 110 films from this database with commercially available soundtracks that include the original motion picture score and purchased these soundtracks as MP3 digital downloads (see S1 Appendix for details).

**Film genre.** We labeled the genres of every film in our 110-film dataset by extracting genre tags from IMDb (imdb.com). Although IMDb lists 24 film genres, we only collect the tags of six genres for this study: Action, Comedy, Drama, Horror, Romance, and Science Fiction (Sci-Fi). This reduced taxonomy is well-attested in previous literature [1, 22–24], and every film in our dataset represents at least one of these genres.

We use multi-label genre tags because many movies span more than one of the genres of interest. Further, we conjecture that these movie soundtracks would combine music that has characteristics from each genre in a label set. Statistics of the data set that we use is given in Table 1.

### Automatically extracting musical cues in film

We developed a methodology we call Score Stamper that automatically identifies and time-stamps musical cues from a soundtrack that are used in its corresponding film. A given track from the soundtrack may be part of multiple cues if clips from that track appear in the film on multiple occasions.

The Score Stamper methodology uses Dejavu's audio fingerprinting tool [25], which is robust to dialogue and sound effects. Default settings were used for all Dejavu parameters. The

**Table 1. A breakdown of the 110 films in our dataset.** Only 33 of the films have only one genre tag; the other 77 films are multi-genre. A list of tags for every movie is given in S1 Appendix.

| Genre Tag | Number of Films |
|---|---|
| Action | 55 |
| Comedy | 37 |
| Drama | 44 |
| Horror | 11 |
| Romance | 13 |
| Science Fiction | 36 |

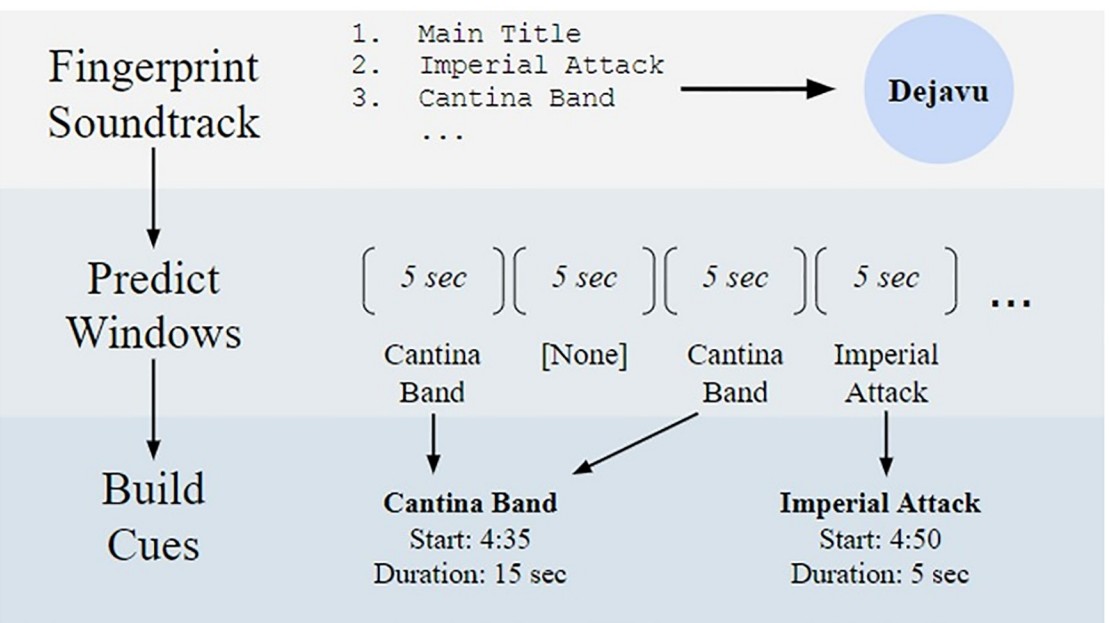

**Fig 1. The Score Stamper pipeline.** A film is partitioned into non-overlapping five-second segments. For every segment, Dejavu will predict if a track in the film's soundtrack is playing. Cues, or instances of a song's use in a film, are built by combining window predictions. In this example, the "Cantina Band" cue lasts for 15 seconds because it was predicted by Dejavu in two nearby windows.

Score Stamper pipeline is explained in Fig 1. At the end of the Score Stamper pipeline, each film has several "cue predictions."

We evaluated Score Stamper's prediction performance on a test set of three films: "The Lion King" (1994), "Love, Actually," and "Star Wars: Episode IV—A New Hope." These films were selected to determine if Dejavu would be robust to markedly different film genres, with different composers, directors, and instrumentation for each set of cues. Additionally, "Love Actually" uses a compilation soundtrack, while the other two films feature composed soundtracks. Musical cues were annotated manually in-house. A total of 84 cues, spanning 162 minutes, were identified across the three films.

Score Stamper's predictions reached an average precision of 0.94 (*SD* = .012) and an average recall of 0.47 (*SD* = .086). We deemed these metrics acceptable for the purposes of this study, as a high precision score indicates that almost every cue prediction Dejavu provides will be correct, given that these test results generalize to the other films in our dataset. The recall is sufficient because the cues recognized are likely the most influential on audience response, as they are included on the commercial soundtrack and mixed clearly over sound effects and dialogue in the film. High recall is made difficult or impossible by several confounding factors: the omission of some songs in a film from its purchasable soundtrack, variations on the soundtrack rendition of the song, and muted placement of songs in the mix of the film's audio.

This result also suggests that Score Stamper overcomes a limitation encountered in previous studies: in prior work, the whole soundtrack was used for analysis (which could be spurious given that soundtrack songs are sometimes not entirely used, or even used at all, in a film) [1, 8, 26]. By contrast, only the music *found in a film* is used in this analysis. Another benefit of this method is a timestamped ordering of every cue, opening up opportunity for more detailed temporal analysis of music in film.

## Musical feature extraction

**MIR features.** Past research in movie genre classification suggests that auditory features related to energy, pitch, and timbre are predictive of film genre [27]. We apply a similar process to [1, 28, 29] in this study: we extract features that relate to dynamics, pitch, rhythm, timbre, and tone using the eponymous functions in MATLAB's MIRtoolbox [30] with default parameters. Spectral flatness and spectral crest are not available in MIRtoolbox, so we compute them using the eponymous functions in Audio Toolbox [31] with default parameters (see Table 2). To capture high-level information and align features extracted at different frequencies, all features are then "texture-windowed" by calculating mean and standard deviation of five-second windows with 33% overlap, as in [32].

**VGGish features.** In addition to the aforementioned features, we also extract embeddings from every cue using VGGish's pretrained model [34]. In this framework, 128 features are extracted from the audio every.96 seconds, which we resample to 1 Hz to align with the MIR features. These embeddings have shown promise in tasks like audio classification [35], music recommendation [36], and movie event detection [37]. We compare the utility of these features with that of the MIR features.

## Visual features

Following previous works in low-level visual analysis of films [14, 15], we extract two features from each film in our dataset: brightness and contrast. These features were sampled at 1 Hz to align with musical features. Brightness and contrast were calculated as in [14], given by:

$$B_t = \frac{1}{|P_t|} \sum_{P_t}^{p} B_t^{(p)} \tag{1}$$

and

$$C = (B_{Max} - B_{Min})(1 - |Area_{B_{max}} - Area_{B_{min}}|) \tag{2}$$

where C is the contrast, $P_t$ is the set of all pixels onscreen at timestep $t$, $B_t^{(p)}$ is the brightness at pixel $p$ at timestep $t$, and $B_{Max}$ and $B_{Min}$ refer to the maximum and minimum average brightness across pixels, evaluated per timestep.

# Methods

## Genre prediction model training procedure

In order to select the model architecture which could best predict film genre from musical features, we use leave-one-out cross-validation, meaning that a model is trained for each of the 110 films in the corpus using the other 109 films. As the ground truth label for each movie can

**Table 2. Auditory features used and feature type.**

| Feature Type | Feature |
|---|---|
| Dynamics | RMS Energy |
| Pitch | Chroma |
| Rhythm | Pulse Clarity [33], Tempo |
| Timbre | MFCCs, ΔMFCCs, ΔΔMFCCs, Roughness, Spectral Centroid, Spectral Crest, Spectral Flatness, Spectral Kurtosis, Spectral Skewness, Spectral Spread, Zero-crossing Rate |
| Tone | Key Mode, Key Strength, Spectral Brightness, Spectral Entropy, Inharmonicity |

contain multiple genres, the problem of predicting associated genres was posed as multi-label classification. For Simple MI and Instance Majority Voting approaches, the multi-label problem is decomposed into training independent models for each genre, in a method called *binary relevance*. The distribution of genre labels is unbalanced, with 55 films receiving the most common label (Action), and only 11 films receiving the least common label (Horror). In order to properly evaluate model performance across all genres, we calculate precision, recall, and F1-score separately for each genre, and then report the macro-average of each metric taken over all genres.

## Model architectures

For the genre prediction task, we compare the performance of several MIL model architectures. First, we explore a Simple MI approach where instances are averaged with one of the following base classifiers: random forest (RF), support vector machine (SVM), or k-nearest neighbors (kNN). Using the same base classifiers, we also report the performance of an instance majority voting approach.

For neural network-based models, the six different pooling functions shown in Table 3 are explored. We adopt the architecture given in Fig 2, which has achieved state-of-the-art performance on sound event detection (SED) tasks [18]. Here, the input feature representation is first passed through three dense embedding layers before going into the pooling mechanism. At the output layer, we convert the soft output to a binary prediction using a fixed threshold of 0.5. A form of weighted binary cross-entropy was used as the loss function, where weights for the binary positive and negative class for each genre are found by using the label distribution for the input training set. An Adam optimizer [38] with a learning rate of 5e-5 was used in training, and the batch size was set to 16. Each model was trained for 200 epochs.

## Frame-level and cue-level features

For each cue predicted by Score Stamper, a sequence of feature vectors grouped into frames is produced (either VGGish feature embeddings or hand-crafted MIR features). For instance, a 10-second cue represented using VGGish features will have a sequence length of 10 and a feature dimension of 128. One way to transform the problem to an MIL-compatible

**Table 3. The six pooling functions, where $x_i$ refers to the embedding vector of instance *i* in a bag set *B* and *k* is a particular element of the output vector h.** In the multi-attention equation, *L* refers to the attended layer and *w* is a learned weight. The attention module outputs are concatenated before being passed to the output layer. In the feature-level attention equation, $q(\cdot)$ is an attention function on a representation of the input features, $u(\cdot)$.

| Function Name | Pooling Function |
|---|---|
| Max pooling | $h_k = \max_i x_{i_k}$ |
| Average pooling | $h_k = \frac{1}{|B|}\sum_i x_{i_k}$ |
| Linear softmax | $h_k = \frac{\sum_i x_{i_k}^2}{\sum_i x_{i_k}}$ |
| Single attention | $h_k = \frac{\sum_i w_i x_{i_k}}{\sum_i w_{i_k}}$ |
| Multi-attention | $h_k^{(L)} = \frac{\sum_i w_{i_k}^{(L)} x_{i_k}^{(L)}}{\sum_i w_{i_k}^{(L)}}$ |
| Feature-level attention | $h_k = \sum_{\mathbf{x} \in B} q(\mathbf{x})_k u(\mathbf{x})_k$ |

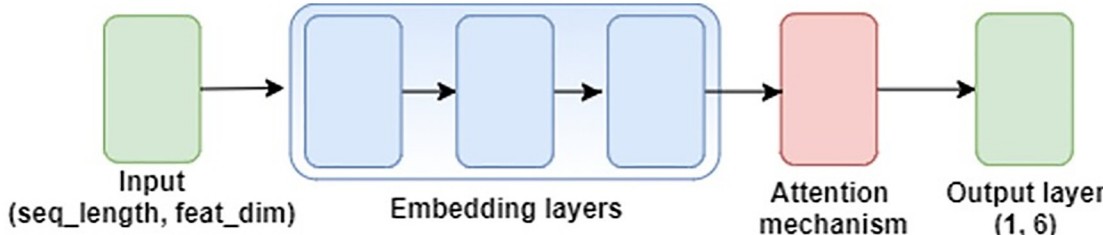

**Fig 2. Neural network model architecture.**

representation is to simply treat all frames for every cue as instances belonging to a movie-level bag, ignoring any ordering of the cues. This approach is called *frame-level* representation.

A simplifying approach is to construct cue-level features by averaging frame-level features per cue, resulting in a single feature vector for each cue. Using MIL terminology, these *cue-level* feature vectors then become the instances belonging to the film, which is a "bag." We evaluate the performance of each model type when frame-level features are used and when cue-level features are used.

## Results

### Genre prediction

Table 4 shows the performance of several model architectures on the 110-film dataset, using either VGGish features or MIR features as input. All of our models outperform both a random guess baseline, using class frequencies, and a zero rule baseline, where the most common (plurality) label set is predicted for all instances. We observe that a previous study, which predicted

**Table 4. Classification results on the 110-film dataset.** Performance metrics using leave-one-out cross-validation for each cue-level feature model are reported. IMV stands for Instance Majority Voting; FL Attn for Feature-Level Attention. Simple MI and IMV results represent performance with the best base classifier (kNN, SVM, and random forest were tried). All models reported mean-averaged precision significantly better than the random guess baseline ($p < .01$), as given by a paired t-test.

| Features | Model | Precision | Recall | F1-Score |
|---|---|---|---|---|
| None | Random Guess | .32 | .32 | .32 |
| | Plurality Label | .14 | .34 | .19 |
| VGGish | kNN—Simple MI | **.64** | .59 | .61 |
| | SVM—IMV | **.64** | .42 | .44 |
| | Max Pooling | .56 | .48 | .49 |
| | Avg. Pooling | .55 | **.78** | .62 |
| | Linear Softmax | .62 | .59 | .59 |
| | Single Attn | .60 | .73 | **.65** |
| | Multi-Attn | .45 | .72 | .52 |
| | FL Attn | .53 | .74 | .57 |
| MIR | SVM—Simple MI | .60 | .52 | .56 |
| | SVM—IMV | .55 | .40 | .42 |
| | Max Pooling | .40 | .10 | .15 |
| | Avg. Pooling | .55 | **.78** | .61 |
| | Linear Softmax | .55 | .61 | .57 |
| | Single Attn | .49 | .76 | .55 |
| | Multi-Attn | .44 | .70 | .51 |
| | FL Attn | .53 | .67 | .56 |

uni-labeled film genres from music tracks, reported a macro F1-score of 0.54 [1]. While important aspects of the two studies differ (track-level vs. film-level prediction, uni-label vs. multi-label genre tags), macro-F1 scores of 0.62 from our best-performing models demonstrate improved performance on an arguably more difficult task.

We note that cue-level feature representations outperform instance-level feature representations across all models, so only values from cue-level feature models are reported. We further observe that Simple MI and IMV approaches perform better in terms of precision, recall, and F1-score when using VGGish features than when using MIR features. This result makes sense, as VGGish embeddings are already both semantically meaningful and compact, allowing for these relatively simple models to produce competitive results. Indeed, we find that Simple MI with an SVM as a base classifier on VGGish features produces the highest precision of all the models we tested. We report precision-recall curves for the top-performing MIR and VGGish models in S2 Appendix. In S3 Appendix, we present a scatter plot with precision and recall for each film (micro-averaged across all cues), for both VGGish and MIR average pooling models.

Finally, we observe that models trained using VGGish features generally outperform their counterparts trained using MIR features. Here, we note that the overall best-performing model in terms of macro-averaged F1-score is a single-attention model with 128 nodes per hidden layer, and trained using VGGish features. Interestingly, pooling mechanisms that are most consistent with the standard MI assumption—Max Pooling and Linear Softmax Pooling [20]—perform worse than other approaches. This result is consistent with the idea that a film's genre is characterized by all the musical cues in totality, and not by a single musical moment.

## Musical feature relevance scoring

To determine the the importance of different musical features toward predicting each film genre, we used the method of Permutation Feature Importance (PFI), as described in [39]. PFI scores the importance of each feature by evaluating how prediction performance degrades after randomly permuting the values of that feature across all validation set examples. The feature importance score $s_k$ for feature $k$ is calculated as:

$$s_k = 1 - \frac{\mathbf{F1}^{perm_k}}{\mathbf{F1}^{orig}} \tag{3}$$

where $\mathbf{F1}^{perm_k}$ is the F1-score of the model across all leave-one-out cross-validation instances with feature $k$ permuted, and $\mathbf{F1}^{orig}$ is the F1-score of the model without any permutations. A high score $s_k$ means that the model's performance degraded heavily when feature $k$ was permuted, indicating that the model relies on that feature to make predictions. This analysis was used to provide an understanding for which features contributed the most to genre predictions, *not* to provide the best-performing model.

To generate the F1-scores, we used our best-performing model trained on MIR features: an average-pooling model with 64 nodes per hidden layer (F1-score = 0.61). We did not analyze a model trained on VGGish features, because VGGish features are not interpretable: a PFI analysis using these features would not illuminate which human-understandable musical qualities contribute most to genre predictions. Since we had a large feature set of 140 features, and many of our features were closely related, we performed PFI on feature *groups* rather than individual features, as in [40]. We evaluated eight feature groups: MFCCs, ΔMFCCs, ΔΔMFCCs, Dynamics, Pitch, Rhythm, Timbre, and Tone. One feature group was created for each feature type in Table 2 (see section "Research data collection and curation"). MFCCs, ΔMFCCs, ΔΔMFCCs were separated from the "Timbre" feature type into their own feature groups, in order to prevent one group from containing a majority of the total features (and

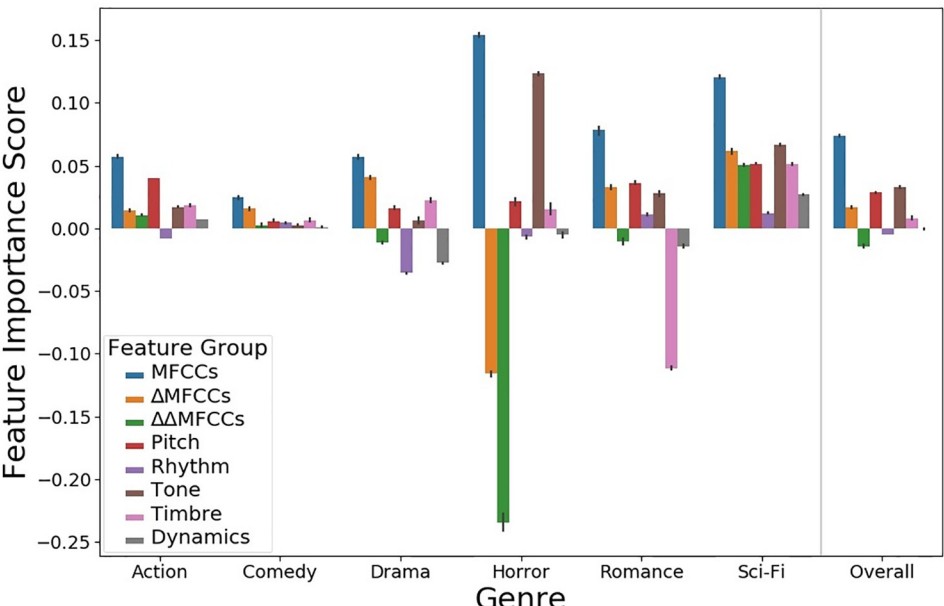

**Fig 3. Feature importance by genre and feature group, reported with 95% CI error bars.**

thus having an overwhelmingly high feature importance score). For each feature group, we randomly permuted all features individually from the others to remove any information encoded in the interactions between those features. We report results averaged over 100 runs in order to account for the effects of randomness. The results of our PFI analysis are shown in Fig 3.

Fig 3 shows that MFCCs were the highest scoring feature group in every genre. Across all genres (i.e., the "Overall" column in Fig 3), the next-highest scoring feature groups were Tone, Pitch, and ΔMFCCs. This corroborates past research finding MFCCs to be the best-performing feature group for various music classification tasks [41, 42]. MFCC and ΔMFCC features were the only ones to significantly degrade performance when permuted for the Comedy and Drama genres, suggesting that those genres may be most characterized by timbral information encoded in MFCCs.

The model relied heavily on the Tone feature group in distinguishing Horror films. Brownrigg [5] qualitatively posits that atonal music or "between-pitch" sounds are characteristic of Horror film music, and the model's reliance on Tone features–including key mode, key strength, spectral brightness, spectral entropy, and inharmonicity–supports this notion. The Tone feature group also scored highly for the Romance and Sci-Fi genres, whose scores often include modulations in texture and key strength or mode to evoke feelings of passion and wonder, respectively.

Finally, we note that the model's predictions for Horror and Romance exhibited greater score variance during feature permutation than for the other genres, likely because Horror and Romance were under-represented in the 110-film corpus.

## Musical-visual cross-modal analysis

To investigate whether visual features associated with a genre correlate with music that the model has learned to be characteristic of that genre, we compare median screen brightness and contrast from film clips with labeled musical cues. For instance: if the model finds that music

from a given film is highly characteristic of Comedy (regardless of the actual genre labels of the film), do we observe visual features in that film that are characteristic of Comedy?

We consider three different sources of genre labels: the true labels, the predicted labels from the best-performing model, and the predicted genre labels where only *false positives* are counted (that is, true positive genre predictions are removed from the set of all genre predictions.) By comparing the brightness and contrast averages using the actual and predicted labels, we can analyze whether musical patterns that the model finds characteristic of each genre correspond to visual patterns typical of the genre.

We use a single-attention pooling model trained on VGGish features (F1-score = 0.65). For each genre, we report the difference between the median brightness or contrast value in film clips labeled with that genre against the median value of all other clips. Table 5 shows the results.

From the "Actual" metrics, we observe that for both brightness and contrast, our dataset follows the trends illustrated in [14]: Comedy and Romance films have high average brightness and contrast, while Horror films have the lowest values for both features. However, we also note that clips from Sci-Fi films in our dataset also have high contrast, which differs from the findings of [14].

When comparing the brightness and contrast of clips by their "Predicted," rather than "Actual," genre, we note that the same general trends are present, but tend more toward the global median for both metrics. This movement toward the median suggests that the musical styles the model has learned to associate with each film genre do not necessarily correspond to their visual styles; e.g., a clip with music befitting Comedy may not keep the Comedy-style visual attributes of high brightness and contrast. This gap is partially explainable by the fact that the model has imperfectly learned the musical differences between genres. However, insofar as the model has learned an approximation of musical characteristics distinguishing film genres, we contend that the difference between the "Actual" and "Predicted" visual averages is

**Table 5. Difference in median brightness and contrast ($\times 10^1$) across all films labeled with a given genre against median brightness and contrast of the set of films excluding the given genre.** Bold values show a statistically significant difference, as given by a Mann-Whitney U test with Bonferroni correction ($\alpha = 0.01$, m = 6) between the median of films including a given genre against those excluding it, within a given prediction source (Actual, Predicted, or False Positive).

| | Brightness | | |
| --- | --- | --- | --- |
| | Actual | Predicted | False Positive |
| **Action** | **0.08** | **0.06** | **-0.18** |
| **Comedy** | **0.23** | **0.19** | **0.16** |
| **Drama** | **-0.07** | **0.01** | **0.31** |
| **Horror** | **-0.70** | **-0.44** | **-0.15** |
| **Romance** | **0.11** | **0.17** | **0.16** |
| **Sci-Fi** | **0.01** | -0.04 | **-0.07** |

| | Contrast | | |
| --- | --- | --- | --- |
| | Actual | Predicted | False Positive |
| **Action** | -0.08 | 0.00 | 0.13 |
| **Comedy** | **0.53** | **0.45** | **0.38** |
| **Drama** | **-0.25** | **-0.16** | **0.31** |
| **Horror** | **-0.28** | **-0.25** | **0.27** |
| **Romance** | **0.35** | **0.16** | **-0.08** |
| **Sci-Fi** | **0.30** | **0.05** | **-0.20** |

an approximation of the difference between visual styles in a film's labeled genre(s) against those genre(s) that its music alone would imply.

To further support this notion, we present the "False Positive" measure, which isolates the correlation between musical genre characteristics and visual features in movies *outside* that genre. For instance, in an Action movie with significant Romance musical characteristics (causing the model to assign a high Romance confidence score), do we observe visual features associated with Romance? For half of the genres' brightness values, and a majority of the genres' contrast values, we actually found the opposite: "False Positive" metrics tended in the opposite direction to the "Actual" metrics. This unexpected result warrants further study, but we suspect that even when musical style subverts genre expectations in a film, the visual style may stay consistent with the genre, causing the observed discrepancies between the two modes.

## Conclusion

In this study, we quantitatively support the notion that characteristic music helps distinguish major film genres. We find that a supervised neural network model with attention pooling produces competitive results for multi-label genre classification. We use the best-performing MIR feature model to show that MFCC and tonal features are most suggestive of differences between genres. Finally, we investigate the interaction between musical and low-level visual features across film genres, but do not find evidence that music characteristic of a genre implies low-level visual features common in that genre. This work has applications in film, music, and multimedia studies.

## Supporting information

**S1 Appendix. Complete list of films used in this study.**
(PDF)

**S2 Appendix. Precision-recall curves for top-performing MIR and VGGish models.**
(PDF)

**S3 Appendix. Scatter plot displaying precision and recall for each film (micro-averaged across all cues), for both VGGish and MIR average pooling models.**
(PDF)

**S1 Data.**
(ZIP)

## Author Contributions

**Conceptualization:** Benjamin Ma.

**Data curation:** Dillon Knox.

**Funding acquisition:** Shrikanth Narayanan.

**Investigation:** Benjamin Ma, Timothy Greer, Dillon Knox.

**Methodology:** Benjamin Ma, Dillon Knox.

**Project administration:** Benjamin Ma.

**Supervision:** Shrikanth Narayanan.

**Visualization:** Timothy Greer.

**Writing – original draft:** Benjamin Ma, Timothy Greer, Dillon Knox.

**Writing – review & editing:** Benjamin Ma, Timothy Greer, Dillon Knox.

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
