## [Decision Letter · Decision Letter 0]

4 Dec 2020

PONE-D-20-31042

A computational lens into how music characterizes genre in film

PLOS ONE

Dear Dr. Ma,

Thank you for submitting your manuscript to PLOS ONE. After careful consideration, we feel that it has merit but does not fully meet PLOS ONE’s publication criteria as it currently stands. Therefore, we invite you to submit a revised version of the manuscript that addresses the points raised during the review process.

We look forward to receiving your revised manuscript.

Kind regards,

Stavros Ntalampiras

Academic Editor

PLOS ONE

Journal Requirements:

"The study was done at the Center for Computational Media Intelligence at USC, which is supported by a research award from Google. The funders had no role in study design, data collection and analysis, decision to publish, or preparation of the manuscript."

We note that you received funding from a commercial source: Google.

Reviewers' comments:

Reviewer's Responses to Questions

**Comments to the Author**

1. Is the manuscript technically sound, and do the data support the conclusions?

Reviewer #1: Yes

Reviewer #2: Yes

Reviewer #3: Yes

2. Has the statistical analysis been performed appropriately and rigorously? 

Reviewer #1: Yes

Reviewer #2: I Don't Know

Reviewer #3: Yes

3. Have the authors made all data underlying the findings in their manuscript fully available?

Reviewer #1: Yes

Reviewer #2: Yes

Reviewer #3: Yes

4. Is the manuscript presented in an intelligible fashion and written in standard English?

Reviewer #1: Yes

Reviewer #2: Yes

Reviewer #3: Yes

5. Review Comments to the Author

Reviewer #1: This work presents interesting and original findings about the use of music genre in films. Authors provide a proper literature review and the language is clear.

=======

Methods

=======

You could provide more details about p-values and statistically significance, but, honestly, I do not mind about this too much. The main issue, however, is that the difference among the models used should be inspected further. One method that I suggest is to show the results (i.e. tables 4 and 5) using violin plots, which allow a general qualitative overview of the distribution without falling in type I and II errors. Moreover, violin plots are easily to build. Another option is to just use a scatter plot in a Precision-Recall space and different colors for different models.

You say that you have used p-value for checking results in table 5, but what test have you used? Why have you chosen that test? Have you corrected it with some method (e.g. Bonferroni/Holm methods)? Have you used a multi-distribution test such as Kruskal-Wallis or ANOVA?

About the features, why have you chosen those features? Which previous studies have you followed? Have you simply used the Matlab standard features? If yes, why?

In figure 3, you show the feature importance plot for MIR features; I did not get why you have not computed the feature importance for the VGG features: reducing the number of features used for a classification model may lead to and increase of the overall performance. Again, why have you not used confidence intervals/violin plots/box plots or similar in this figure? It is hard to understand what is the importance of each feature otherwise.

Ease of reading

You should also declare more precisely the contribute of their work in the abstract and possibly in the introduction: the sentence that is at now is almost unuseful.

To my understanding, the paragraph "Visual-musical cross-modal analysis" has almost no reason to be. You repeat everything later, while previous works should be put in "Related works". Note that an extnsive survey about multi-modal and cross-modal music studies extists [1].

Paragraph "Multiple Instance Learning" is very unclear. You should say as soon as possible what is a bag and what is an instance in your study. Everything should then be referenced to your case (e.g. hypothesis etc.). This allows the reader to understand MPI with a concrete example.

Results reported about ScoreStamper in paragraph "Automatically extracting musical cues in film" are scientifically unuseful. You have tested it in only 3 bags. How many instances, stamps, multiple occasions there were? How much were differente the music pieces in the song track? Reasons about the low recall are unclear.

You should describe with more details the structure of the models used, even if these models were already used in previous papers.

=========

References

=========

[1] F. Simonetta, S. Ntalampiras, and F. Avanzini, “Multimodal Music Information Processing and Retrieval: Survey and Future Challenges,” in Proceedings of 2019 International Workshop on Multilayer Music Representation and Processing, Milan, Italy, 2019, pp. 10--18. https://arxiv.org/abs/1902.05347

Reviewer #2: This paper presents a study on the use of films' soundtracks to help automatic classification of their broad genre (e.g. Action, Comedy). To do so, the authors 1) curated a new dataset of films and their corresponding sound tracks with automatic fingerprinting annotations; 2) explored different features for describing the music content of soundtracks, in particular they explored common MIR features related to dynamics, rhythm, pitch, timbre and tone (such as MFCCs, tempo and chroma) and deep learning derived features (VGGish); 3) investigated different variations classifiers and problem definitions (SVMs, kNNs, DNNs, Multiple Instance Learning, different pooling strategies); 4) performed an ablation study on the importance of the MIR features for the classification of the different genres and its relation with simple image clues such as brightness and contrast.

The paper is well written, has good references to previous work and a thoughtful discussion of the results, so I'd recommend it to be accepted. Even though the paper is in good shape already, there is room for improvement in the structure and more details about implementation and experiments should be added. See my comments below.

Improvement in structure ===

In my opinion the contributions should be better highlighted. It is a bit difficult to understand what exactly the contributions are with respect to previous work since the discussion in related work doesn't clearly highlight the limitations in the literature besides some isolated comments spread out in the text (i.e. the first clear statement I saw was in line 46). The innovations in methodology, and the analysis should be also listed as contributions if they're more extensive than previous works.

More detail on the experiments ===

Authors should explain how they assessed statistical significance. It is mentioned in the text but not clarified. Also it would be good to have more details on the network architecture besides a reference (capacity or number of parameters, layer's size, input size a bit more clear). Note that the dataset is not very big and this raises some questions on suitability of the architecture that could be partially answer with more information on the implementation.

I think some reference to the performance of previous works is needed to understand if the models presented here are performing reasonably (which they seem when I went compared to [1]) but an explicit comment would help understand the work better.

I'm not sure if I followed the conclusions in L298-L302 that the music style of a clip. The conclusion is that because the brightness and contrast in the clips using predicted labels are not correlating with the "expected behaviour" of each genre then the music in the clip doesn't necessarily correspond to the visual style? Then why do you see that effect in the clips when you use "actual labels"? Could it be an artifact of the model's performance?

Nit comments ===

- The figures are not in the main text, not sure if this is an artifact of the reviewing template or something to correct.

- L154: Briefly explain "texture windowed"

- L166: Would prefer a short explanation on how brightness and contrast were obtained and refer to paper for further details.

- L221: I don't understand this phrase, couldn't parse it.

- L283 - L286: Are you trying to say that investigating brightness and contrast mean scores on the model's predictions help understand associations between those visual features and what the model learned? And that could potentially be applied to unknown genres? Maybe rephrase to make it clearer.

Reviewer #3: The authors build on research in the field, utilizing well-constructed computational models to retrieve information and data to help us understand how film music operates across several genres and interacts with other film modalities. They have applied this to over 100 films, and I find that their approach in identifying the music from the soundtracks that are actually timestamped in the film itself is a sound and even essential one. Though this study does involve necessary technical information appropriate for a study like this, they are careful to take the reader step-by-step through their process, carefully explaining terms, and leading us to their conclusions in a logical, cogent manner. They have also provided strong data in support of the study. I believe that this study can open the way to further research, as they even suggest in the paper. As a practicing musician and musical scholar myself, I will be interested in seeing this lead to further published work that will help us better understand music’s role as significant and interactive cinematic device, and how viewers respond to the cinematic experience, emotively, perceptively, and cognitively.

6. PLOS authors have the option to publish the peer review history of their article (what does this mean?). If published, this will include your full peer review and any attached files.

Reviewer #1: No

Reviewer #2: No

Reviewer #3: **Yes: **Joseph L. Rivers Jr.

---

## [Author Response · Author response to Decision Letter 0]

25 Jan 2021

(This response is also available in the Response to Reviewers file upload.)

To the academic editor and reviewers:

Thank you for your insightful comments on our manuscript submission. We have reviewed your comments one-by-one and prepared a revised document with changes based on your feedback. In this letter, we describe each reviewer comment and the changes we have made to address it.

We received editorial feedback to amend our Competing Interests statement to acknowledge this study’s partial funding from a corporation, Google. We have updated our Cover Letter to include an amended Competing Interests statement. Please let us know if other clarifications must be made to the manuscript in this regard.

We received feedback from reviewers advising us to provide more details about p-values and statistical significance of our results. To this end, we conducted Mann-Whitney U tests on our results from Table 5 and corrected them with Bonferroni correction.

To show that our results were replicable, we re-ran all of our experiments with leave-one-out cross-validation. We believe that this further strengthens the conclusions drawn from the experimental modeling results. At the behest of one of the reviewers, we also include scatter plots in the Precision-Recall space to better illustrate the flexibility and performance of our models.

We hope that the statistical tests that we have implemented in the revised submission will help further substantiate our experimental findings, and the conclusions drawn from our work.

One reviewer asked us to justify our choice of music information retrieval (MIR) features, as well as clarify how we calculated them. The MIR features we chose have shown utility in music emotion recognition and music genre classification from prior published work. Default parameters were used for computation and then texture-windowed to provide our models feature sets with equal window lengths. In the manuscript, we have clarified how we computed our MIR features, and cited prior works which inspired our choice for the features used in our study. Our revised submission now includes specific details of which previous studies we were following and how we computed the features we used for our study.

We thank the reviewers for inquiring why the feature importance analysis was not conducted for VGGish features. We have added a sentence in our manuscript that mentions that VGGish features are not interpretable like MIR features are (for the reason that VGGish features do not correlate with human-understandable musical descriptors, such as loudness), so a feature importance plot for VGGish features would similarly not be interpretable. Our motivation for performing PFI was not to improve model performance, but rather to get a sense for which features contributed most to genre predictions. We have amended our manuscript to address this comment. Finally, we have added confidence interval indicators to Figure 3, which displays the results of the PFI analysis.

The reviewers advised us to more explicitly state the contributions and potential applications of our work. To address this feedback, we have added in a new section to our manuscript called “Contribution of this Work,” which we hope will allow the reader to more precisely understand what we add to this area of study.

We appreciate the reviewers for pointing out a useful reference for cross-modal studies involving music. We have included the citation in our updated manuscript. Additionally, we have trimmed the “Visual-musical cross-modal analysis” section to avoid redundancy in the results section.

We received feedback that the “Multiple instance learning” sub-section in Related Works is unclear. In our most recent revision, we feel that the section is better motivated and clearer with regard to what a bag and an instance are in the context of our study. We believe the manuscript now better reflects why our problem could be framed as an MIL task, and we thank the reviewer for pointing out that this will allow the readers of our manuscript to better understand multiple instance learning and its relationship to this study.

A reviewer asked for more details on Score Stamper’s performance on three films that were manually annotated in-house. We have added to our manuscript the number of instances per movie, as well as our justification for choosing the three movies (they are of different genres, by different directors, and contain different style soundtracks).

At the request of the reviewers, we have added more information about the models and architectures that we used in our study. Additionally, we have expanded our study to include leave-one-out error analysis, which we hope provides more rigor to our study, especially given the relatively small size of our dataset. In our new manuscript, we more clearly show our method’s efficacy in predicting film genres and give more details about our models so that our study may be more easily reproduced. Finally, we have added a comparative analysis of our models’ performance against the baseline classifiers and prior studies.

Thank you for the comments about the need for additional clarity in the conclusions on “Musical-visual cross-modal analysis”. We have now added remarks on the various implications of our results. When we compare visual and musical styles by genre, the musical styles in question are the characteristics the model has learned to associate with every genre, which are not necessarily the actual musical characteristics that distinguish different film genres. Just as the model has learned an approximation of musical characteristics distinguishing film genres, our analysis approximates the relationship between visual and musical characteristics in different film genres.

Our goal in our musical-visual cross-modal analysis is to determine if visual features associated with a genre correlate with music that the model has learned to be characteristic of that same genre. We have updated the manuscript to clarify this motivation for readers.

We thank our reviewers for their interest in our manuscript and improving it. We hope that our work will find some use to our readers and others who are interested in understanding music’s role as a significant and interactive cinematic device.

---

## [Decision Letter · Decision Letter 1]

17 Feb 2021

PONE-D-20-31042R1

A computational lens into how music characterizes genre in film

PLOS ONE

Dear Dr. Ma,

Thank you for submitting your manuscript to PLOS ONE. After careful consideration, we feel that it has merit but does not fully meet PLOS ONE’s publication criteria as it currently stands. Therefore, we invite you to submit a revised version of the manuscript that addresses the points raised during the review process.

We look forward to receiving your revised manuscript.

Kind regards,

Stavros Ntalampiras

Academic Editor

PLOS ONE

Reviewers' comments:

Reviewer's Responses to Questions

**Comments to the Author**

1. If the authors have adequately addressed your comments raised in a previous round of review and you feel that this manuscript is now acceptable for publication, you may indicate that here to bypass the “Comments to the Author” section, enter your conflict of interest statement in the “Confidential to Editor” section, and submit your "Accept" recommendation.

Reviewer #1: (No Response)

Reviewer #2: All comments have been addressed

2. Is the manuscript technically sound, and do the data support the conclusions?

Reviewer #1: Partly

Reviewer #2: (No Response)

3. Has the statistical analysis been performed appropriately and rigorously? 

Reviewer #1: No

Reviewer #2: Yes

4. Have the authors made all data underlying the findings in their manuscript fully available?

Reviewer #1: Yes

Reviewer #2: Yes

5. Is the manuscript presented in an intelligible fashion and written in standard English?

Reviewer #1: Yes

Reviewer #2: Yes

6. Review Comments to the Author

Reviewer #1: 1) Statistical significance tests were only performed for the visual feature experiments and not for the VGG-ish and MIR features.

2) the caption of table 5 is unclear about what the bold style means: authors say to have computed p-values using bonferroni correction; this means that they have compared multiple tests, but I cannot understand what are these tests: did they compare all the tests for which they show the average at once? or did they compared "actual", "predicted" and "false positives" in each row? or all rows of "predicted" and then all rows of "actual" and then all rows of "false positives"?

3) Moreover, statistical tests need the same cardinality between the set tested. This would mean that the number of "films labeled with a given genre" is the same as the number of "films excluding the given genre", which seems unlikely. How they managed this problem for the statistical test?

4) I appreciate the addition of supplementary figure S2, but it shows recision-recall curves, not scatter plots. PR-curves comes out when the classification is made based on some threshold, and they are a method for evaluating the model, not the distributions of the predictions. AUC (Area under curve) is similar to F1-score and in this case PR-curves don't add any knowledge. Even if other reviewers find this plot useful, it's not clear what "no skill" line is. It would be more clear if MIR and VGG-ish points were on the same plot.

When I wrote "scatter plot in a precision-recall space", I was meaning a scatter plot, not a curve. Scatter plots are plots which shows points in their coordinates; in my example, coordinates were precision and recall. Points could be, for instance, each film. Multiple distributions can be plotted using different colors, eg. multiple models: see for instance the following image https://www.researchgate.net/profile/Mohit_Bansal7/publication/267783907/figure/fig3/AS:669377024258060@1536603327607/BCubed-Precision-Recall-scatter-plot-for-the-Japanese-English-dataset-Each-point.png A plot such that could be useful to qualitatively evaluate the difference models without falling in type I and type II errors.

Reviewer #2: In my previous review I mentioned that the paper was in good shape already but I recommended the authors to 1) clarify their contributions and structure; 2) provide more details in experiments and discussion, i.e. explain details in architecture, statistical significance, how the visual features were calculated, among others.

The authors addressed all my comments and also improved the results and discussion section which is much more clear now, so I recommend the paper to be accepted in this new version.

7. PLOS authors have the option to publish the peer review history of their article (what does this mean?). If published, this will include your full peer review and any attached files.

Reviewer #1: No

Reviewer #2: No

---

## [Author Response · Author response to Decision Letter 1]

5 Mar 2021

(This response is also available in the Response to Reviewers file upload.)

We thank the reviewers for their feedback, which have helped us further strengthen our manuscript. Below we provide details of how we addressed their comments.

1) Statistical significance tests were only performed for the visual feature experiments and not for the VGG-ish and MIR features.

Response: We overlooked showing statistical significance tests for the experiments involving VGGish and MIR features in the previous manuscript. We have included a paired t-test on the mean average precision (mAP) of our models, where the null hypothesis is that the mAP of our proposed model is not shown to be significantly higher than the random guess baseline. Perhaps not surprisingly, our best-performing models show significantly better performance than the baseline at the 0.01 level. The table has been updated with more details, and we hope the readers will see that these models are significantly better-performing than our baselines.

2) The caption of table 5 is unclear about what the bold style means: authors say to have computed p-values using bonferroni correction; this means that they have compared multiple tests, but I cannot understand what are these tests: did they compare all the tests for which they show the average at once? or did they compared "actual", "predicted" and "false positives" in each row? or all rows of "predicted" and then all rows of "actual" and then all rows of "false positives"?

Response: We appreciate the feedback about our caption in table 5: we have clarified this caption to indicate that we were using 6 hypotheses in our Bonferroni Correction, which correspond to comparing the median of a particular genre in a particular column (“Actual,” “Predicted,” and “False Positives”) with the median of all of the other genres in that same column. Our caption has been reworded and proper parameters are included to elucidate the tests that we ran.

3) Moreover, statistical tests need the same cardinality between the set tested. This would mean that the number of "films labeled with a given genre" is the same as the number of "films excluding the given genre", which seems unlikely. How they managed this problem for the statistical test?

Response: While there are some statistical tests that need the same cardinality between the two samples tested, we used a Mann Whitney U test in our results reported in Table 5, which does not mandate that the number of films labeled with a given genre is the same as the number of films excluding the given genre.

4) I appreciate the addition of supplementary figure S2, but it shows precision-recall curves, not scatter plots. PR-curves comes out when the classification is made based on some threshold, and they are a method for evaluating the model, not the distributions of the predictions. AUC (Area under curve) is similar to F1-score and in this case PR-curves don't add any knowledge. Even if other reviewers find this plot useful, it's not clear what "no skill" line is. It would be more clear if MIR and VGG-ish points were on the same plot. When I wrote "scatter plot in a precision-recall space", I was meaning a scatter plot, not a curve. Scatter plots are plots which shows points in their coordinates; in my example, coordinates were precision and recall. Points could be, for instance, each film. Multiple distributions can be plotted using different colors, eg. multiple models: see for instance the following image https://www.researchgate.net/profile/Mohit_Bansal7/publication/267783907/figure/fig3/AS:669377024258060@1536603327607/BCubed-Precision-Recall-scatter-plot-for-the-Japanese-English-dataset-Each-point.png A plot such that could be useful to qualitatively evaluate the difference models without falling in type I and type II errors.

Response: We appreciate our reviewer’s clarification about the PR plot. We have amended our manuscript to include models based on VGGish features and MIR features on the same PR plot to facilitate easy comparison of models, as was shown in the reviewer’s example. Each film’s cues were compiled and the predictions of these cues were used to calculate micro-averaged precision and recall scores for a film. We label the highest- and the lowest-precision and recall films, which we believe indicates that the performance between models using MIR features and VGGish features is similar for each film. We hope this new figure complements the PR curves shown in supplementary figure S2 by allowing readers to easily compare between MIR and VGGish-based models.

---

## [Decision Letter · Decision Letter 2]

29 Mar 2021

A computational lens into how music characterizes genre in film

PONE-D-20-31042R2

Dear Dr. Ma,

We’re pleased to inform you that your manuscript has been judged scientifically suitable for publication and will be formally accepted for publication once it meets all outstanding technical requirements.

Kind regards,

Stavros Ntalampiras

Academic Editor

PLOS ONE

Additional Editor Comments (optional):

Reviewers' comments:

Reviewer's Responses to Questions

**Comments to the Author**

1. If the authors have adequately addressed your comments raised in a previous round of review and you feel that this manuscript is now acceptable for publication, you may indicate that here to bypass the “Comments to the Author” section, enter your conflict of interest statement in the “Confidential to Editor” section, and submit your "Accept" recommendation.

Reviewer #1: All comments have been addressed

Reviewer #2: All comments have been addressed

2. Is the manuscript technically sound, and do the data support the conclusions?

Reviewer #1: Yes

Reviewer #2: Yes

3. Has the statistical analysis been performed appropriately and rigorously? 

Reviewer #1: (No Response)

Reviewer #2: Yes

4. Have the authors made all data underlying the findings in their manuscript fully available?

Reviewer #1: Yes

Reviewer #2: Yes

5. Is the manuscript presented in an intelligible fashion and written in standard English?

Reviewer #1: Yes

Reviewer #2: Yes

6. Review Comments to the Author

Reviewer #1: All the comments have finally been addressed. Statistical significance tests have been carried out over all the evaluations shown in the text and an additional plot has been added, giving the rough idea about how different the two models are. Since PLOS One guidelines instruct the reviewers to stress the rigorousity of the scientific procedure in respect to the results, even if the plot added shows how VGG-ish features only marginally outcomes classic MIR features, I find that the paper still adds knowledge worth of publication.

Reviewer #2: The authors addressed my comments previously, and they have carefully answered and made the changes requested for the other reviewer in this iteration. The paper is in good shape to me.

7. PLOS authors have the option to publish the peer review history of their article (what does this mean?). If published, this will include your full peer review and any attached files.

Reviewer #1: No

Reviewer #2: No

---

## [Editor Report · Acceptance letter]

31 Mar 2021

PONE-D-20-31042R2 

A computational lens into how music characterizes genre in film 

Dear Dr. Ma:

I'm pleased to inform you that your manuscript has been deemed suitable for publication in PLOS ONE. Congratulations! Your manuscript is now with our production department. 

Kind regards, 

on behalf of

Prof. Stavros Ntalampiras 

Academic Editor

PLOS ONE